# EEG-Based Evidence of Mirror Neuron Activity from App-Mediated Stroke Patient Observation

**DOI:** 10.3390/medicina57090979

**Published:** 2021-09-17

**Authors:** Jin-Cheol Kim, Hyun-Min Lee

**Affiliations:** 1Department of Physical Therapy, City Hospital, Seomun-daero 654, Gwangju 61710, Korea; kjcbboy@gmail.com; 2Department of Physical Therapy, College of Health Science, Honam University, Honamdae-gil 100, Gwangju 62399, Korea

**Keywords:** EEG, mirror neuron system activity, stroke, StrokeCare application, activities of daily living

## Abstract

*Background and Objectives*: The mirror neuron system in the sensorimotor region of the cerebral cortex is equally activated during both action observation and execution. Action observation training mimics the functioning of the mirror neuron system, requiring patients to watch and imitate the actions necessary to perform activities of daily living. StrokeCare is a user-friendly application based on the principles of action observation training, designed to assist people recovering from stroke. Therefore, when observing the daily life behavior provided in the StrokeCare app, whether the MNS is activated and mu inhibition appears. *Materials and Methods*: We performed electroencephalography (EEG) on 24 patients with chronic stroke (infarction: 11, hemorrhage: 13) during tasks closely related to daily activities, such as dressing, undressing, and walking. The StrokeCare app provided action videos for patients to watch. Landscape imagery observation facilitated comparison among tasks. We analyzed the mu rhythm from the C3, CZ, and C4 regions and calculated the mean log ratios for comparison of mu suppression values. *Results:* The EEG mu power log ratios were significantly suppressed during action observation in dressing, undressing, walking, and landscape conditions, in decreasing order. However, there were no significant activity differences in the C3, C4 and CZ regions. The dressing task showed maximum suppression after a color spectrum was used to map the relative power values of the mu rhythm for each task. *Conclusions:* These findings reveal that the human mirror neuron system was more strongly activated during observation of actions closely related to daily life activities than landscape images.

## 1. Introduction

The mirror neuron system (MNS) consists of a distributed network of neurons with mirror-like properties, such that activation occurs in cerebral motor regions during observation of movement, behavior, language, etc. [1]. Mirror neurons are activated upon the observation of various motion-related movements, such as sit-to-stand, walking, dressing, drinking, and reaching. In addition, they become specifically activated during the observation of desired actions [2] or even by goal-oriented movements [3].

First detected in the human ventral premotor cortex (VPC), mirror neurons have also been found in the inferior frontal gyrus (IFG) and superior temporal sulcus (STS) [4]. Currently, the human MNS is known to include the posterior IFG, adjacent VPC, and rostral part of the inferior parietal lobule (IPL) [5].

Electroencephalography (EEG) is a non-invasive recording technique which measures electrical neural currents at scalp level. Neural oscillations can have different frequency bands: 0–4 Hz (delta), 3–7 Hz (theta), 8–12 Hz (alpha), 13–30 Hz (beta), 20–32 Hz (high beta), and 30–50 Hz (gamma). EEG oscillations allowed the identification of the mu rhythm (8–13 Hz) associated with MNS activation [6]. In general, the mu rhythm is activated in the sensorimotor cortex at rest and suppressed during the observation of movement [7]. There is a positive correlation between MNS activation and mu rhythm suppression [8]. Therefore, the mu rhythm can be considered an index for sensorimotor area modulation by mirror neuron activity [9]. As such, sensorimotor area activity is believed to generate the mu rhythm that leads to mirror neuron activation in several cortical regions [10]. A previous study reported significant mu rhythm suppression in the sensorimotor cortex during the observation of a movement sequence consisting of rest, flat hand gesture, grip formation, and grip execution [11]. These results suggest that observing can activate the sensorimotor cortex and affect motion processing.

In another action observation study investigating the mu rhythm, the observed tasks and movements activated and changed the associated brain regions [12]. Previous studies focused on observing motions, such as biting an apple, grasping a cup or a ball, kicking a ball, and pushing a brake [13]. Those studies showed that the ventral parts of Brodmann areas (BA) 6, 44 and 45 were activated upon the observation of a motion toward the mouth. In contrast, the dorsal parts of BA6 and BA45 were activated upon the observation of hand movements, such as gripping and grasping. Foot-related movements resulted in the activation of the dorsal part of BA6 [14]. Thus, the level of motion or type of activity of a body part during action observation facilitates the inference of the associated brain region, including C3, C4, or Cz. Arnstein et al. used EEG and Functional Magnetic Resonance Imaging (fMRI) to measure the brain activation during motion observation to determine MNS activation [15]. They found that the mu rhythm in the EEG was significantly inhibited during action observation. Pineda et al. [16] used EEG and fMRI to match mirror neuron areas, finding activation of the BA44, primary motor sensory cortex (BA2), dorsal premotor cortex (DPM), supplementary motor cortex (SMA), and IPL, suggesting the involvement of the MNS region and mu rhythm suppression during the observation of an action.

The purpose of this study was to observe MNS activation and mu rhythm in the cerebral cortex using the StrokeCare application (app) during action observation training as well as to present a new paradigm of neurorehabilitation of patients with stroke (infarction: 11, hemorrhage: 13).

## 2. Materials and Methods

### 2.1. Participants

Twenty-six patients with chronic stroke were recruited from rehabilitation hospitals in Gwangju Metropolitan City. Participants gave their written informed consent before the study initiation, after being informed of the aim and methods. This study was approved by the Bioethics Committee of the Honam University (Registration number: 1041223-201705-HR-05). The sample number of subjects in this study was set based on the significance level of 0.05 required for repeated ANOVA using the G*power 3.1 programs, the effect size of 0.4, the power of 0.80 according to Cohen’s law, and a total of 2 variables. The minimum number of samples was 23, so this study was found to satisfy the appropriate number of samples [17].

Among those 26 patients, 24 were included in the study (Figure 1). One patient with a comorbid neurological disease was excluded, and another deemed unsuitable for EEG measurement because the patient was scheduled to undergo a cranial extraction operation.

We selected participants according to the following inclusion and exclusion criteria:

#### 2.1.1. Inclusion Criteria

Definite diagnosis of cerebrovascular disease;A stroke >6 months ago, before the onset of the study;Korean-Mini-Mental Status Examination (MMSE-K) score ≥24 without any cognitive impairment and ability to follow instructions;Capable of independent movement using an auxiliary tool with a Motor Assessment Scale score ≥20;No auditory or visual impairment or aphasia;No neglect symptoms.

#### 2.1.2. Exclusion Criteria

Other neurological diseases;Craniectomy;Orthopedic diseases of the lower extremities.

### 2.2. Experimental Procedure

#### App and Audiovisual Material

The StrokeCare app was developed by Professor Hyun-Min Lee of the Department of Physical Therapy at Honam University, with the aim of facilitating daily life movements in patients with nervous system damage. StrokeCare is an app developed with the support of the National Research Foundation of Korea. The research project is the development and usability evaluation of motion observation training-based daily life motion education for stroke patients (2017R1C1B5017978). The participants observed the images and videos recorded in the app using a 10.1-inch tablet (Galaxy tap A, Samsung, Korea) in a sound-proofed room. To exclude the possibility that the subject’s attention may momentarily decrease or the influx of miscellaneous waves during EEG measurement, the baseline and late lines at the beginning and end of the EEG were configured with a white background (Figure 2a,f). The audiovisual material consisted of male and female actors dressing (Figure 2b), undressing (Figure 2c), and walking at a comfortable speed over 10 m (Figure 2d). Pictures of landscapes, such as mountains, oceans, waterfalls, beaches, and gardens were added to establish a point of comparison (Figure 2e). The videos were prepared as follows: baseline (15 s), task monitoring (40 s), and late-stage conditions (15 s). A white screen was shown during baseline and late-stage conditions. The total duration of each condition was 1 min.

### 2.3. EEG Data Acquisition

We attached electrodes to the participants’ scalp using the international 10–20 electrode placement method. Next, we conducted the EEG recording while the subject was observing each task for 1 min.

We used the BIOS-S24 (BioBrain Inc., Daejeon, Korea) model for the EEG. We recorded the subjects’ brainwaves on a computer using a 256 Hz sampling frequency, 0.5–50 Hz bandpass filter, and 12-bit Analog Digital (AD) conversion. We collected EEG data using 19 electrodes attached to each subject’s scalp. The electrodes used were Fp1, Fp2, F7, F3, FZ, F4, F8, T3, C3, CZ, C4, T4, T5, P3, PZ, P4, T6, O1 and O2. In addition, we attached reference electrodes to the right ear and at the back of the neck (Figure 3). Each EEG electrode was a flat gold-coated disk electrode. While attaching each electrode, we removed foreign substances from the head surface with a cotton swab dipped in alcohol. This was performed to minimize contact resistance with the skin. In addition, we used an electrode paste (Z401CE, Laxtha, Korea), and fixed the electrodes by covering them with gauze to prevent them from falling off.

After attaching the electrodes, the participants sat in a comfortable chair in a quiet examination room, where they were asked to watch the videos on the tablet PC provided. We questioned the patients informally about the contents of the videos after recording the EEG, to gauge their level of attention.

### 2.4. EEG Data Analysis

We collected EEG data for each condition by recording the signals over 70 s. We removed the first 5 s of EEG corresponding to the beginning and end of each video, to exclude the possibility of an instantaneous drop in attention or jitter. We digitally converted the data and analyzed it using the BioScan program (BioBrain Inc., Daejeon, Korea). As mu rhythm suppression occurs most reliably in the sensory motor cortex, we analyzed the data obtained from C3, CZ and C4 locations more closely corresponding to the neural areas of mirror neuron activation.

We calculated mu inhibition by converting the mu power ratio of each condition to that at baseline. A log ratio < 0 indicates a suppression in the mu rhythm, while a ratio > 0 indicates an increase. A value corresponding to 0 suggests no suppression. We recorded mu suppression in the contralateral hemisphere to the hand.

We investigated the main effects of the hemisphere on data acquired from C3 and C4 sites, as well as their interactions when participants were watching each of the dressing, undressing, walking, and landscape videos.

All data were analyzed using SPSS (SPSS 21.0 for Windows, SPSS Inc, Chicago, IL, USA). To determine the general characteristics of subjects, the average and standard deviation were expressed through descriptive statistics. The Kolmogorov–Smirnov test was used to analyze the normal distribution of the population. Moreover, a two-way analysis of variance was used to determine the main effect of the hemisphere, as well as the interaction between hemisphere and condition. We performed post-hoc tests using the Bonferroni method for multiple comparisons if the main effect was found to be significant. We tested whether there was a significant difference in mu suppression at C3 and C4 for each condition using an independent *t*-test. This study implemented the statistical procedure used by Babiloni et al. [18]. The level for statistical significance was set to 0.05.

## 3. Results

### 3.1. General Characteristics of EEG Subjects

Among the 24 patients with stroke who underwent EEG measurements, 17 were men and seven women. Right-sided paresis, left-sided paresis, cerebral infarction, and brain hemorrhage were observed in 13, 11, 11 and 13 cases, respectively. The average disease duration was 52.08 ± 30.27 months, and the mean age 59.95 ± 8.54 years. The average MMSE-K score was 26.66 ± 2.21 (Table 1).

### 3.2. Average Mu Power Log Ratio during Observation of the Motion

Table 2 shows the mu power log ratio corresponding to the baseline.

### 3.3. Average Mu Power Log Ratio According to Brain Hemisphere

An ANOVA model with repeated measurements of condition (dressing, undressing, walking, landscape) X brain hemisphere (right, central sphere, left) mixed factor was used to investigate the change in mu rhythm. The average of the log ratios of mu power in the C3, CZ, and C4 regions was compared by repeated-measured two-factor ANOVA. The main effect of the condition was significant (df = 3, F = 163.97, *p* < 0.001) but differences between hemispheres was not. Furthermore, there was a substantial difference in the interaction of the brain with the video conditions. We performed repeated measurement variance analysis for each brain hemisphere. Post-hoc analysis results are presented in Figure 4.

According to the post-hoc analysis, the mu power log ratio corresponding to the dressing movements was significantly lower in the left hemisphere (C3) than those of undressing (*p* < 0.001), walking (*p* < 0.001), and landscape (*p* < 0.001). The mu power log ratio at the central electrode (CZ) observed during dressing was significantly lower than that during walking (*p* < 0.001) and landscape (*p* < 0.001) watching. However, there was no significant difference in the log ratio observed during undressing (*p* = n.s.). In addition, the measurement on the right hemisphere (C4) during dressing was substantially lower than that of undressing (*p* < 0.001), walking (*p* < 0.001), and landscape (*p* < 0.001).

### 3.4. Mu Power Log Ratio for Each Task

We calculated the average of the relative power log ratios and compared them to determine the difference in mu power log ratios between different tasks. The average mu power log ratio of C3, CZ and C4 for the dressing condition was −0.031 ± 145, while that for the undressing condition was −0.019 ± 0.006. The ratios for the walking and landscape conditions were −0.007 ± 0.007 and −0.004 ± 0.001, respectively (Table 3).

We conducted a repeated measures analysis of variance to compare the ratio under each condition. We found a main effect of condition (df = 3, F = 100.603, *p* < 0.001) and the post-hoc analysis showed a significantly higher mu suppression ratio under the dressing condition than under the undressing (*p* < 0.001), walking (*p* < 0.001), and landscape (*p* < 0.001) watching conditions.

### 3.5. Mu Rhythm Mapping for Each Condition

Figure 4 shows an EEG intensity map showing the change in mu rhythm under each condition. The size of the relative power value of the mu rhythm under each condition is represented continuously by a color spectrum. The relative power value decreases toward purple. A strong purple intensity for each condition suggests mu rhythm suppression (Figure 5).

## 4. Discussion

The sensorimotor cortex is not only active during motion performance, but also while observing the movements and actions of others, through MNS activation [19]. The MNS is strongly triggered in the VPC, IPL and STS [20]. The mu rhythm obtained from EEG is used as an indicator of MNS activation. This is thought to be spontaneously triggered by sensory motor nerves at rest. However, it is suppressed upon observation of the actual movement. This facilitates the determination of the sensorimotor region modulation home to the MNS. A previous study reported that frequencies ranging from 8 to 13 Hz recorded at the C3, CZ and C4 electrodes were accompanied by large amplitude EEG waveforms [21]. In this study, we investigated the mu rhythms recorded from C3, CZ and C4 sites when participants were watching dressing, undressing, and walking actions, or landscape videos.

As we recorded the difference in mu inhibition in the right and left-brain hemispheres and compared mu rhythm log ratios (according to the central spheres of patients with stroke under each condition), mu suppression at C3 was significant in the order of undressing, walking, and landscape, based on dressing. It was suppressed and significantly different under each condition. In contrast, mu suppression at CZ was significantly suppressed compared to dressing for undressing, walking, and landscape, in decreasing order, while no substantial difference between dressing and undressing was observed. Furthermore, the mu rhythm at C4 was significantly suppressed under the undressing, walking, and dressing conditions, and the difference was significant between dressing and each of the remaining conditions.

Franceschini et al. [22] measured mu rhythms at C3 and C4 sites during the observation of object-related pictures, and non-object-related and object-related conditions. Mu suppression was more significant upon observing object-related conditions than the remaining two. In addition, MNS activity was identified by the mu rhythms during the observation of hand movements [23]. In addition, we also recorded it in the ipsilateral hemisphere during observation of the moving hand and it was reportedly higher than inhibition. A previous study examined mu suppression in both hemispheres during the observation of motion and reported the lack of significant difference between the cerebral hemispheres [24]. In this study, during the observation of dressing and undressing actions, greater mu inhibition was recorded at the left and right electrodes than at the central site. However, there was no significant difference between the two hemispheres. These results are consistent with those of previous studies. Thus, mu rhythm activation in both cerebral hemispheres can be accredited to the functional activity involving the use of both hands while putting on and removing clothes.

Calvo-Merino et al. examined brain activation by observing the movements of individuals performing traditional ballet dance and Capoeira, a Brazilian traditional martial art form [25]. The MNS, including the VPC, parietal cortex, and STS, were strongly activated. Researchers also found activation in the anterior region of BA6 through actions that involved kicking a real object, applying a brake, and a similar action using a foot without an object [26]. Furthermore, mu suppression was higher at the central electrode site than in either hemisphere during observation of the gait conditions. In addition, the swinging motion of the arm resulted in the activation of both hemispheres. However, it is presumed that the leg movements substantially activated the central sulcus, which lies in a deeper region.

Previous studies have also reported differences in mu inhibition by comparing MNS activation corresponding to stimulation conditions [27]. A study on children with autism also reported no mu rhythm suppression with unfamiliar motions, despite being a familiar task [28]. In another study, researchers observed different brain regions during piano playing by professional players and the general public. MNS activation was higher in the former than in the latter. Thus, MNS activity varies depending on the conditions of the observed stimulus [29].

Brain imaging studies have reported the association of the MNS with other brain areas, including the occipital lobe, temporal lobe, and parietal cortex, all of which play a role in vision [30]. Action observation and performance seem to activate the motor system in a similar manner. Furthermore, action observation and performing a movement during observation provide similar advantages [31]. Furthermore, action observation may affect the execution of a specific motion. Therefore, motor cortex projection areas may be involved in observation-execution matching during the performance of certain target-related actions.

The primary purpose of imitation is to match the observed motion with the learned motor repertoire of the observer. In addition, observation of new movement patterns helps identify the nerve regions associated with a particular area. While learning a new movement pattern, the corresponding motion projection areas in the IPL and VPC in the MNS are classified and integrated for each movement. The MNS modulates learning by recombining and circulating to re-fit the observed model. This leads to the activation of human sensory information.

Previous studies that examined MNS by action observation used desktop monitors and laptop screens. However, we also observed changes in MNS using an app on a 10-inch tablet PC in our study. This may help overcome the limitations of the existing rehabilitation programs.

This study has limitations in generalizing the results due to the small sample size of the experimental group. In addition, it was difficult to compare the results with healthy people because this study design only investigated mu inhibition for stroke patients without a control group. The low spatial resolution of the EEG prevents direct confirmation of MNS activation. We indirectly investigated MNS activation through EEG recordings in the mu frequency band. Future studies may focus on technologies such as fMRI and positron emission tomography (PET), which have higher spatial resolution than EEG and may facilitate the identification of the activated regions in the MNS.

## 5. Conclusions

The study findings suggest that observation of daily activities such as dressing, undressing, and walking activate the MNS as indicated by mu rhythm suppression. The rehabilitation of movements required for performing daily life activities in stroke patients may thus be aided by action observation, suggesting the possibility of extending the scope of the StrokeCare application for time-saving educational and training purposes. Future research should include patients with stroke who have not actually participated in an expensive rehabilitation environment to compare action observations in various environments and places in the community.

## Figures and Tables

**Figure 1 medicina-57-00979-f001:**
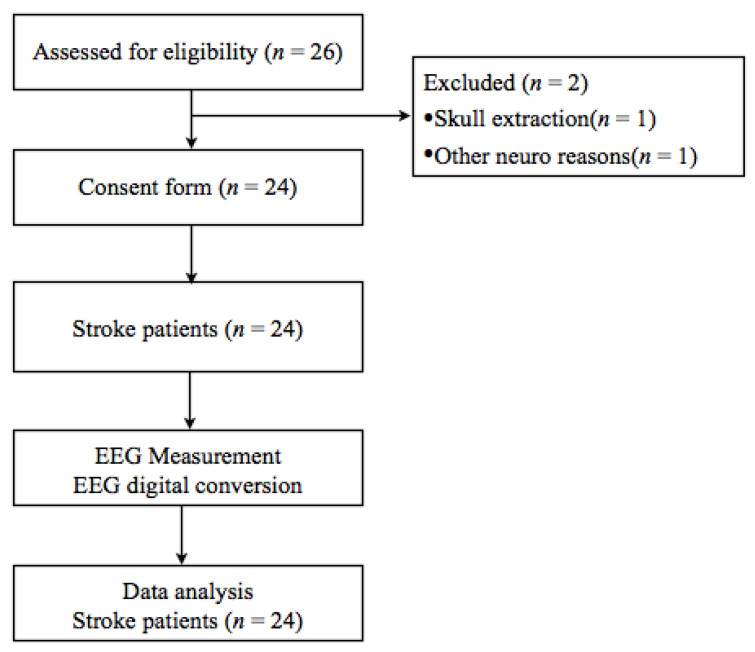
Electroencephalography (EEG) Experimental procedure.

**Figure 2 medicina-57-00979-f002:**
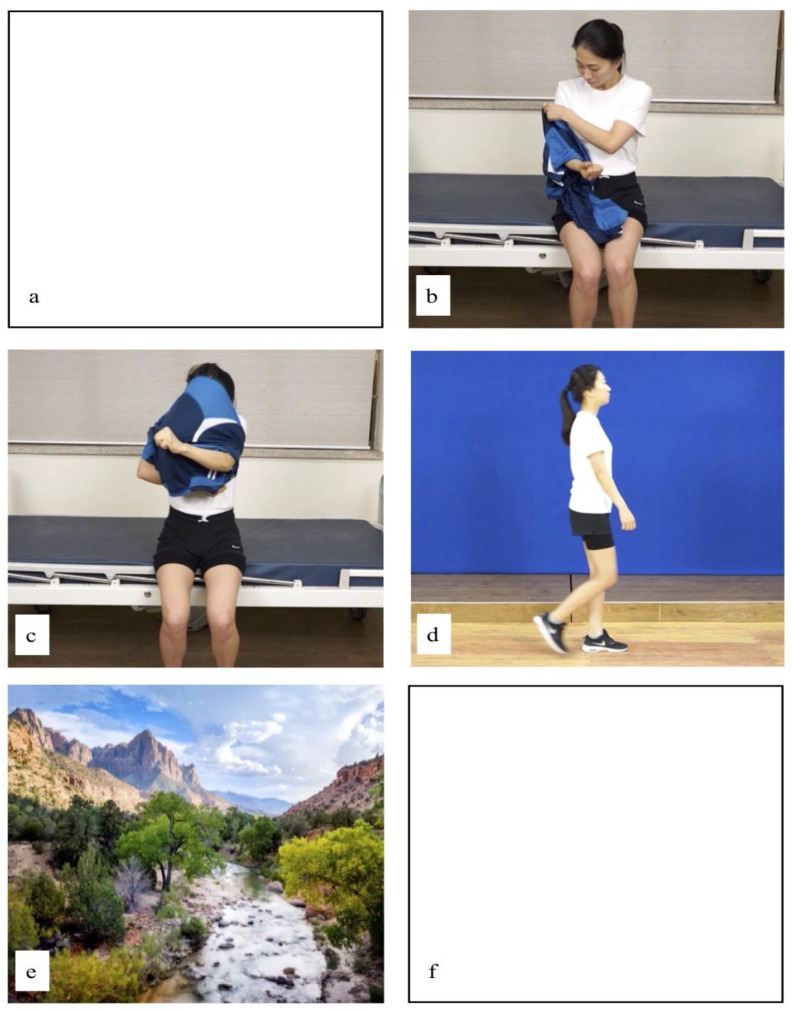
Action and landscape imagery conditions. ((**a**). baseline, (**b**). dressing, (**c**). undressing, (**d**). gait, (**e**). landscape imagery, and (**f**). post-baseline).

**Figure 3 medicina-57-00979-f003:**
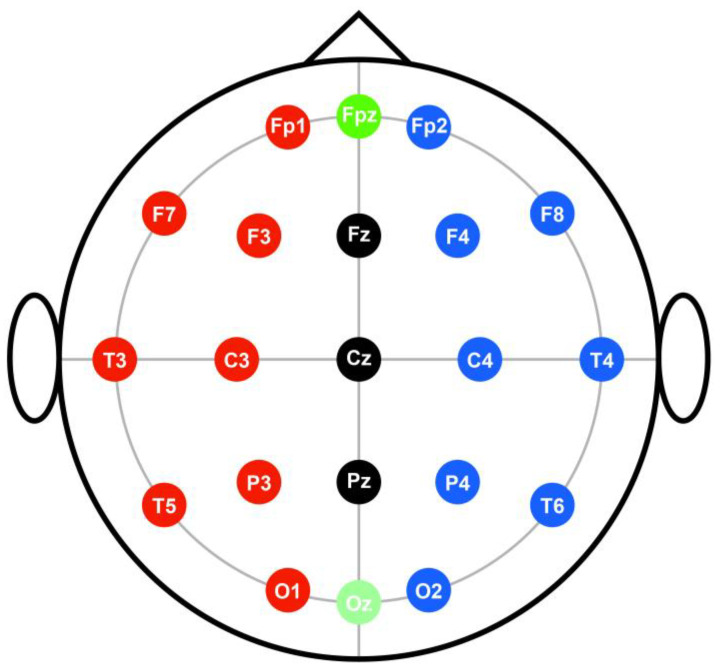
Electroencephalography (EEG) international 10–20 method. Electrodes were attached to Fp1, Fp2, F7, F3, FZ, F4, F8, T3, C3, CZ, C4, T4, T5, P3, PZ, P4, T6, O1, O2.

**Figure 4 medicina-57-00979-f004:**
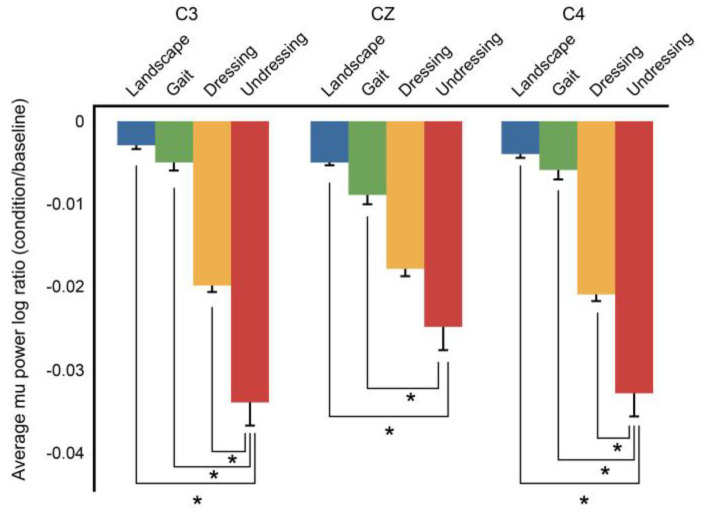
C3, CZ, C4 of average mu power log ratio. Mu inhibition concerning landscape, gait, dressing and undressing conditions of each hemisphere C3, CZ and C4 is indicated. The error bars for each condition represent the standard error of the mean. Significant inhibition is shown by (* *p* < 0.05).

**Figure 5 medicina-57-00979-f005:**
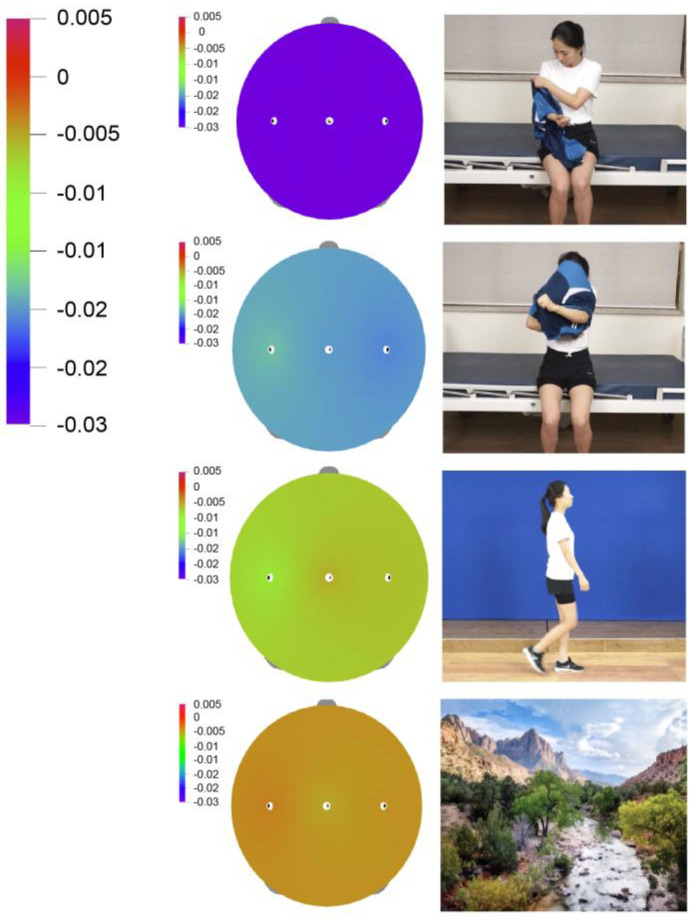
Mu rhythm mapping change. Electroencephalography (EEG)-based brain mapping indicated cortical activation during action observation training.

**Table 1 medicina-57-00979-t001:** Patient demographics.

	Characteristics	Stroke Patients (*n* = 24)
Sex	Male	17
	Female	7
Paretic side	Left	11
	Right	13
Stroke type	Infarction	11
	Hemorrhage	13
Onset period (months)		52.08 ± 30.27 *
Age (years)		56.95 ± 8.54
MMSE-K		26.66 ± 2.21 **

* Mean ± standard deviation. ** Korean Version of Mini-Mental State Examination; MMSE-K.

**Table 2 medicina-57-00979-t002:** Average mu power log ratio of action observation.

	Stroke Patients (*n* = 24)		
	C3	CZ	C4
Dressing	−0.034 ± 0.009 *	−0.025 ± 0.008	−0.033 ± 0.005
Undressing	−0.020 ± 0.012	−0.018 ± 0.008	−0.021 ± 0.008
Gait	−0.005 ± 0.002	−0.009 ± 0.018	−0.006 ± 0.002
Landscape imagery	−0.003 ± 0.002	−0.005 ± 0.003	−0.004 ± 0.002

* Mean ± standard deviation. C3; left hemisphere, CZ; central sphere, C4; right hemisphere.

**Table 3 medicina-57-00979-t003:** Average mu power log ratio of C3, CZ and C4.

	Stroke Patients
Dressing	−0.031 ± 0.145
Undressing	−0.019 ± 0.006
Gait	−0.007 ± 0.007
Landscape imagery	−0.004 ± 0.001 *

* Mean ± standard deviation.

## Data Availability

No new data were created or analyzed in this study. Data sharing is not applicable to this article.

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
