# Peer review of "EEG-Based Evidence of Mirror Neuron Activity from App-Mediated Stroke Patient Observation"

_medicina, 2021, doi:10.3390/medicina57090979_

Round 1

Reviewer 1 Report

This study was aimed at observing changes in the mirror neuron system and mu rhythm in the cerebral cortex in response to action observation training in stroke patients. Here are my comments:

Major comments:

1) The authors have conducted EEG analysis during exposure to a series of pictures on stroke patients. I think it is equally important to do the same in controls (healthy individuals) so we know what is the baseline. 

2) Was the StrokeCare app developed in this study? If so then it needs more description. If not then it needs proper citation.

3) The sample size is very less for the authors to make the claim. They should write that as a limitation. 

4) The discussion is very disjointed and not very coherent. Their results are lost among the results from previous studies. The main highlights of their results don't come through.

5) The authors should define mu rhythm. It is an important concept to know to understand the results. It might be an oversight by the authors, but having the definition will orient the readers.

6) The figure legends need to be more descriptive. Currently, it is just a title.

7)  There are many grammatical errors throughout the paper that make reading difficult.

Minor comments:

1) Line 44 EEG recordings allow not allowed

2) Line 74, remove by app-based action observation training. It is repetitive.

3) Line 179 should read between hemispheres were not.

4) The sentences on Line 238 and line 239 both start with In addition.  It doesn't make sense since one talks about a previous study and the other about this study.

Author Response

Dear reviewer,

I carefully read the reviewer's suggestions and made the corrections. The modifications are summarized as follows.

Major comments:

1) This study is a quasi-experimental design study without a control group. We plan to conduct further expanded studies targeting the experimental group and the control group in the future.

2) Added content to the Experimental procedure.

3) I added the limitations of the study in the last paragraph of the discussion.

4) The second paragraph of the review provides a general summary of the findings of this study. Significant differences are explained when the scenery, gait, dressing, and undressing conditions are observed in each cerebral hemisphere. I think this partly explains the highlights of the findings of this study sufficiently.

5) In the third paragraph of the introduction, the mu rhythm is defined.

6) I supplemented figure legends.

7) I wrote a revision last week and requested an English correction. I will submit the revision I wrote first, and once the submission is confirmed, I will submit the file that has been corrected in English.

Minor comments:

1) I edited the recordings into oscillations.

2) I deleted the repetitive part.

3) I have corrected that.

4) I deleted the sentences line 238 and line 239.

5) References 11 to 16 in the introductory section contain information on biological mechanisms and aids in the recovery of stroke patients. References 23-26 also referred to studies in which the MNS was activated when conditions related to daily life were observed. This evidence summarize all of the reviewer's suggestions. Figure 2-f image has been modified.

Reviewer 2 Report

The current manuscript is an interesting study showing that the human mirror neuron system is more activated when patients watch videos that are closely related to daily life activities than landscape images. However, there are also several issues the authors should address as summarized below:

  1. The authors should add some contents in introduction/discussion section about why watching the videos can assist patients recovering from stroke and what’s the biological mechanism?
  2. Have the authors checked patients’ MNS activation when they watch other daily activities (e.g. toothbrushing, playing sports)? If not, the authors should at least add some discussion about other daily activities, not just dressing, undressing, and walking, as mentioned in the manuscript.
  3. Figure 2f is a blank image. The authors should double check it.

Author Response

Dear reviewer,

I carefully read the reviewer's suggestions and made the corrections. The modifications are summarized as follows.

1) References 11 to 16 in the introductory section contain information on biological mechanisms and aids in the recovery of stroke patients.

2) References 23-26 also referred to studies in which the MNS was activated when conditions related to daily life were observed. This evidence summarize all of the reviewer's suggestions.

3) Figure 2-f image has been modified.

Round 2

Reviewer 1 Report

The authors have addressed all the comments raised.

Author Response

Dear reviewer,

I corrected the problem raised by the reviewer as follows.

Point 1:

The abstract does not have any aim formulated.

Response 1: 

I added a sentence about aim to abstract. 

Point 2: 

Have the authors calculated the sample size needed for the study to be carried out? The sample size in this study is very small. Adequate statistical power is most important when the results of the study are not statistical significant. An underpowered study is negative but inconclusive: it does not support associations, but neither can it rule out one out.

Response 2: 

I used G*power to calculate the number of subjects and corrected the text.

The sample number of subjects in this study was set based on the significance level of .05 required for repeated ANOVA using the G*power 3.1 programs, the effect size of .4, the power of .80 according to Cohen's law, and a total of 2 variables. The minimum number of samples was 23, so this study was found to satisfy the appropriate number of samples.

Point 3: 

Figures 2a and f are not cited. Please cite them in the text and clearly indicate if they have to be blank.

Response 3: 

I added explanations for figures 2a and 2f.

To exclude the possibility that the subject's attention may momentarily decrease or the influx of miscellaneous waves during EEG measurement, the baseline and late lines at the beginning and end of the EEG were configured with a white background (Figure 2a, 2f).

Point 4: 

The authors state that "The main effect of the condition was significant (df = 3, F = 163.97, p<0.001) but differences between hemispheres not". It is difficult to understand this sentence as ANOVA shows a significant difference in general, but multiple comparisons procedure did not reveal any differences between the groups? Please clarify and provide more precise interpretation.

Response 4: 

II have revised and improved the description of statistical methods.

An ANOVA model with repeated measurements of condition (dressing, undressing, walking, landscape) X brain hemisphere (right, central sphere, left) mixed factor was used to investigate the change in mu rhythm.

Point 5: 

In the footnotes of Figure 4, the authors state that "Significant inhibition is shown by (*p<0.05, **p<0.01)"; however, no ** (two asterisks) are depicted in this figure. Please clarify.

Response 5: 

I removed **p<0.01 from the figure legend.

Point 6: 

Based on the reference list of this manuscript, only 3 cited articles are less than 5 years old. Is this topic relevant and worth to be investigated?

Response 6: 

The StrokeCare app used in this study was developed with the support of the National Research Foundation of Korea in 2016. Our study developed an app to summarize meaningful activities in stroke patients' daily life activities. Since it is a study to prove the effect, the number of previous papers to refer to was small. In addition, it is judged that the study's value will be sufficient compared with the study conducted with the existing simple action observation training video.